# Proteomic Analysis of Methylglyoxal Modifications Reveals Susceptibility of Glycolytic Enzymes to Dicarbonyl Stress

**DOI:** 10.3390/ijms23073689

**Published:** 2022-03-28

**Authors:** Leigh Donnellan, Clifford Young, Bradley S. Simpson, Mitchell Acland, Varinderpal S. Dhillon, Maurizio Costabile, Michael Fenech, Peter Hoffmann, Permal Deo

**Affiliations:** 1Clinical and Health Sciences, Health and Biomedical Innovation, University of South Australia, Adelaide 5000, Australia; donly017@mymail.unisa.edu.au (L.D.); bradley.simpson@unisa.edu.au (B.S.S.); varinderpal.dhillon@unisa.edu.au (V.S.D.); maurizio.costabile@unisa.edu.au (M.C.); michael.fenech@unisa.edu.au (M.F.); 2Clinical and Health Sciences, University of South Australia, Adelaide 5000, Australia; clifford.young@unisa.edu.au (C.Y.); mitch.acland@gmail.com (M.A.); peter.hoffmann@unisa.edu.au (P.H.); 3Centre for Cancer Biology and SA Pathology, University of South Australia, Frome Road, Adelaide 5000, Australia; 4Genome Health Foundation, North Brighton, Adelaide 5048, Australia

**Keywords:** methylglyoxal, glycation, post-translational modifications, MG-H1, CEL, CEA, proteomics, glycolysis

## Abstract

Methylglyoxal (MGO) is a highly reactive cellular metabolite that glycates lysine and arginine residues to form post-translational modifications known as advanced glycation end products. Because of their low abundance and low stoichiometry, few studies have reported their occurrence and site-specific locations in proteins. Proteomic analysis of WIL2-NS B lymphoblastoid cells in the absence and presence of exogenous MGO was conducted to investigate the extent of MGO modifications. We found over 500 MGO modified proteins, revealing an over-representation of these modifications on many glycolytic enzymes, as well as ribosomal and spliceosome proteins. Moreover, MGO modifications were observed on the active site residues of glycolytic enzymes that could alter their activity. We similarly observed modification of glycolytic enzymes across several epithelial cell lines and peripheral blood lymphocytes, with modification of fructose bisphosphate aldolase being observed in all samples. These results indicate that glycolytic proteins could be particularly prone to the formation of MGO adducts.

## 1. Introduction

Post-translational modifications (PTMs) of proteins play an important role in biological events by regulating protein activity, interactions, stability and degradation [1]. PTMs can be derived through enzymatic processes via specific proteins responsible for the addition or removal of modifications including phosphorylation, methylation, acetylation and ubiquitination [1]. Moreover, PTMs can also be derived from non-enzymatic covalent modifications caused by electrophilic metabolites which react with the nucleophilic groups of arginine and lysine residues [2]. These electrophilic metabolites include acetyl-CoA, malonyl-CoA, 4-hydroxynonenal, glyoxal and methylglyoxal (MGO) [2,3]. MGO is a highly reactive 1,2-dicarbonyl compound formed as a by-product of glycolysis, in which intermediate triosephosphates (dihydroxyacetone phosphate and glyceraldehyde-3 phosphate) spontaneously degrade to generate MGO [4]. Because MGO is formed via glucose metabolism, it has been implicated in the pathogenesis of diabetic complications [5]. Moreover, MGO has been shown to be elevated in cancer cells following the metabolic rewiring from oxidative phosphorylation to glycolysis (Warburg effect) as their primary energy source [6]. Cellular MGO concentrations are maintained by the glyoxalase system. MGO spontaneously reacts with glutathione, forming a hemithioacetal that is recognized by glyoxalase 1 (GLO1) and converted to S-D-lactoylglutathione. This is subsequently recognized by GLO2 and converted to the non-reactive D-lactate and glutathione [7]. MGO that is not sequestered by the glyoxalase system is able to react non-enzymatically to form methylglyoxal-hydroimidazolone (MG-H) isomers (1, 2 and 3) on arginine residues and carboxyethyl modifications on both arginine (*N^ε^*-carboxyethylarginine (CEA)) and lysine (*N^ε^*-carboxyethyllysine (CEL)) residues, amongst other less abundant modifications [8]. Collectively, these modifications, as well as those formed by other metabolites (glucose, glyoxal and 3-deoxyglucosone) are referred to as Advanced Glycation Endproducts (AGEs) [9].

Detection of MGO modifications has been largely achieved by measuring the total cellular contribution, rather than its presence on specific proteins [10]. This has primarily been performed utilising spectroscopic, immunochemical, chromatographic and mass spectrometric methods [11]. While these approaches are effective for detecting changes in the abundance of the modification which can be used to infer biological significance, it lacks the specificity to directly identify target proteins, which is crucial for elucidating underlying mechanisms. Unfortunately, measuring these modifications on specific proteins has proved challenging, due to their low abundance, low stoichiometry and heterogeneity, which makes their detection and characterisation difficult. Only a few studies have attempted to measure site-specific MGO modifications in mammalian cells with varying degrees of success [12,13,14,15,16]. Initial studies struggled to identify MGO modifications under control conditions, requiring the use of exogenous MGO to increase the concentration of these modifications [12,13,14,16]. While this is an effective approach for increasing the concentration of these modifications, these conditions may promote non-specific MGO modification of proteins or residues that would otherwise be unmodified under physiological or pathological conditions. Only recently has a comprehensive assessment of MGO modifications in HEK293T cells been applied. In this study, subcellular fractionation was applied to decrease sample complexity allowing for the identification of more than 600 MG-H modified proteins [14].

We recently applied bottom-up proteomics of whole WIL2-NS cell lysates to identify MGO modifications on mitotic proteins to gain a deeper understanding of the aneugenic activity of MGO (manuscript in preparation). While we were able to identify several mitotic proteins harbouring MGO modifications, we also identified an abundance of non-mitotic modified proteins. Therefore, using the entire inventory of MGO modified proteins identified from our previous study, we performed several types of enrichment analysis to identify proteins and/or pathways that are most susceptible to MGO. Samples were digested with ProAlanase (ProAla) or trypsin and analysed by LC-MS. In WIL2-NS cells, we identified over 500 proteins with MGO modifications. Enrichment analysis revealed glycolytic enzymes are targets for MGO modifications, several of which occur on residues involved in substrate interactions and are likely to affect their activity. Furthermore, we showed that the modification of glycolytic enzymes was not unique to WIL2-NS, but was also present in several epithelial cell lines and peripheral blood lymphocytes in vivo, with modification of fructose bisphosphate aldolase being the most prevalent.

## 2. Results

### 2.1. Characterization of MGO Modified Proteins

To explore the potential impact of dicarbonyl stress (elevated MGO) on cellular function, we used a discovery proteomics approach on whole-cell extracts from WIL2-NS cells to investigate the modification of intracellular proteins following treatment with MGO. Cells were treated with 500 µmol/L MGO for 24 h, which led to a 2.2- and 1.8-fold increase in MG-H1 and CEL, respectively, as shown by LC-MS analysis (Appendix A). This relative increase is consistent with those observed in pathological conditions such as diabetes and thus less likely to cause non-specific modifications [8]. 

Preparation of whole-cell extracts for proteomics was performed using trypsin and ProAla in separate digests. Overall, we identified 519 modified proteins (Appendix A). The majority of modifications were identified in trypsin digests; however, the incorporation of ProAla in this study increased the number of identified MGO modification sites by approximately 25% (Appendix A). Surprisingly, only 3.8% of the modification sites were detected in both ProAla and trypsin digested samples (Appendix A). This was mostly due to ProAla modified peptides contributing to increased sequence coverage where the corresponding modification sites were undetected by trypsin; for example, as shown for CEL modification of K20 of Transgelin-2 and K496 of RNA-splicing ligase RtcB homolog (Figure 1A).

Abundant y and b ions were observed for MGO-modified peptides produced in trypsin digested samples, which was particularly useful for identifying site-specific PTM residues (Figure 1B). Moreover, despite a C-terminus alanine or proline, abundant b and y ions could also be observed in ProAla digested samples (Figure 1C). Therefore, ProAla can be used in conjunction with trypsin to increase the identification rate of MGO modifications. To gain insight into what factors may influence the identification of MGO-modified proteins, we ranked the proteins from each digest based on their Log2 abundance. In both digests, modification of proteins by MGO was more often observed in higher abundant proteins, with 45.8% and 61.9% of identifications occurring in the top 20% most abundant proteins for trypsin and ProAla, respectively (Figure 1D and Appendix A). In the trypsin digest, 13 of the 20 most abundant proteins contained MGO modifications, consisting of 32 modification sites. A number of modifications per protein ranged from 1–19 sites, with beta-actin (ACTB) containing the highest number of MGO modification sites (Appendix A).

### 2.2. Enrichment Analysis of MGO Modified Proteins

To explore the functional impact of MGO modifications in WIL2-NS cells, we performed enrichment analysis of biological processes, with the DAVID gene ontology tool using all MGO-modified proteins generated in this study. MGO-modified proteins showed over-representation in a diverse range of biological processes, including canonical glycolysis, translation initiation, regulation of cellular response to heat, mRNA splicing via spliceosome, amongst others (Table 1). Canonical glycolysis was the most enriched biological process (13.4-fold enrichment). Pathways with the largest number of MGO-modified proteins included cell-cell adhesion, mRNA splicing via spliceosome, translation initiation and protein folding, which all contained more than 20 modified proteins (Table 1). To further explore those pathways most impacted by MGO, we developed a protein–protein interaction network with STRING. KEGG pathway enrichment analysis was performed to determine pathways associated with the remaining clusters, which revealed glycolysis/gluconeogenesis, spliceosome and ribosome as the pathways most affected by MGO modifications (Figure 2). Glycolysis involves the conversion of glucose to pyruvate in ten enzyme-mediated steps. Remarkably, seven of the ten steps contained MGO-modified proteins, including glucose-6-phosphate isomerase (GPI), enolase 1/3 (ENO1/3), fructose bisphosphate aldolase A/C (ALDOA/C), triosephosphate isomerase (TPI), glyceraldehyde 3-phosphate dehydrogenase (GAPDH), phosphoglycerate kinase (PGK) and pyruvate kinase M (PKM) consisting of 20 modification sites in total (Figure 3A and Appendix A). The selective enrichment of glycolysis suggests MGO may modulate glucose metabolism by modifying enzymes involved in this pathway. To investigate the potential impact of MGO modifications on these proteins, we looked at the role of residues that were modified. ALDOA is a highly conserved glycolytic enzyme that catalyses the reaction that cleaves the aldol fructose 1,6-bisphosphate into triosephosphates dihydroxyacetone phosphate (DHAP) and glyceraldehyde 3-phosphate (G3P) [17]. Fructose 1,6-bisphosphate covalently binds to K229 followed by Schiff-base formation and carbon-carbon cleavage yielding triosephosphate formation [17]. K229, as well as K146 and R42, which non-covalently interact with the substrate in the active site, were modified by MGO, suggesting modification of these sites is likely to interfere with its interaction with fructose 1,6-bisphosphate (Figure 3B). All four modification sites observed on ALDOA were observed within the active site, despite the presence of numerous other lysine and arginine residing in the protein, suggesting modification of the active site residues was not random (Figure 3B). Strikingly, ENO1, another glycolytic enzyme that converts 2-phosphoglycerate to phosphoenolpyruvate [18], was shown to contain seven modification sites, three of which occurred on residues known to contain other PTMs (Figure 3C).

### 2.3. Modification of Glycolytic Enzymes

Following the identification of glycolytic enzymes as primary targets for MGO modification in WIL2-NS cells, we examined if this was consistent across other cell types. Therefore, using previous data of epithelial cell lines, OV90 and Caov3 without (parental) and with chemoresistance to the chemotherapeutic drug carboplatin (CBPR) from our laboratory, raw data files were reprocessed for MGO modifications. As these cell lines were initially used for a different study, they were not treated with MGO and, therefore, modification sites represent those present under basal conditions. All cell lines, except for OV90 (parental), showed canonical glycolysis to be over-represented with MGO modifications (Figure 4A). Additionally, peripheral blood lymphocytes (PBL) isolated from three separate healthy male donors were similarly analysed for the occurrence of MGO modifications in vivo. As with the epithelial cell lines, analysis of MGO modification sites in PBL was performed without MGO treatment to detect modifications sites present under basal conditions only. Unfortunately, no significant enrichment was observed for MGO modifications on glycolytic enzymes in PBL, but modification of ALDOA was observed in PBL from all donors (Figure 4A,B). Furthermore, modification of ALDOA was observed in all epithelial cell lines (Figure 4B). We also investigated the modification of other proteins that occurred across multiple cell types, Histone H1.1 R57 (CEA), AT-rich interactive domain-containing protein 2 (Fragment) R277 (CEA) and Neuroblast differentiation-associated protein AHNAK K1333 (CEL) being the most commonly identified modification sites (Appendix A).

## 3. Discussion

Metabolite-driven PTMs have a diverse role in regulating protein function, by altering enzyme activity, obstructing protein–protein interactions and by forming covalent crosslinks between proteins [1]. To gain further insight into the role of PTMs, mapping of proteins and specific sites using proteomic techniques is an attractive approach. Therefore, we utilised LC-MS to explore intracellular MGO modifications. Detection of MGO modifications and other AGEs has remained difficult due to their low abundance, low stoichiometry and heterogeneity at modification sites and the lack of suitable enrichment techniques [22]. An ideal approach would utilise pan-specific antibodies to enrich native modifications at the peptide level. Only one previous study has reported this approach for MGO modifications, using antibodies specific to each MG-H isomer and CEA to enrich MGO modified histone peptides; however, this yielded no enrichment [16]. Interestingly, an enrichment method using commercially available antibodies was applied to the AGE *N*^ε^-carboxymethyllysine, where a 10-fold increase in the identification of CML-modified peptides was observed [23]. Several commercially available antibodies exist for MGO modifications and potentially allow a more comprehensive picture of these modifications to be obtained. An alternative enrichment technique used an alkyne-labelled MGO analogue (alkMGO), which can be enriched by click-chemistry based techniques [24]. In this study, 71 MG-H, CEL and CEA modifications were identified in human serum incubated with 250 µmol/L of alkMGO and more than 300 identified in erythrocyte lysates treated with 500 µmol/L [24]. Unfortunately, the alkMGO analogue is not commercially available which limits its use by other researchers. The most common approach to increase the identification of MGO modified proteins is to increase their concentration using exogenous MGO, which was performed in our study. This approach has been used to study MGO modification sites in Human microvascular endothelial cells (HMEC-1), periodontal ligament fibroblasts (PDLF) and HEK293T cells [12,13,14]. In PDLF, only five proteins with MGO modifications were identified under control conditions [13]. This total increased to 172 proteins (consisting of 353 sites) following incubation of the cell lysate with 500 µM MGO for 24 h. Furthermore, the same authors recently provided a comprehensive assessment of the dicarbonyl proteome in HEK293T cells by incubation with 131 µmol/L of MGO followed by subcellular fractionation, which resulted in the identification of more than 600 MG-H modified proteins [14]. Similarly, Galligan et.al (2018) used exogenous MGO in GLO1 KO cells to study the site-specific MGO modifications on histone proteins, where they identified 21 sites across canonical histone proteins [16]. While the addition of exogenous MGO is useful for increasing the concentration and number of MGO modifications, several of the aforementioned studies have used concentrations likely to cause the non-specific modification of proteins beyond physiological or pathological conditions. To prevent excessive modification of proteins, we ensured the concentrations of MG-H1 and CEL following treatment was elevated by less than three-fold to reflect in vivo pathological conditions and minimise the occurrence of non-specific modifications [25]. We also utilised the non-standard protease ProAla, in addition to trypsin to increase total sequence coverage. ProAla has recently been shown to be more effective than trypsin at peptide mapping and identifying PTMs on more basic proteins such as histones, which are known targets for MGO [16,26]. Previous studies have also utilised other proteases, such as Lys-C and Glu-C, for the identification of various glycated proteins [12,23,24]. Therefore, the use of other proteases conjunction with trypsin may be useful in identifying additional MGO modification sites [27].

Enrichment analysis of our inventory of MGO modified proteins using DAVID and STRING revealed distinct networks which were enriched with these modifications. Most enriched was canonical glycolysis, where seven steps out of the ten-step process contained MGO modified proteins in WIL2-NS cells. Previous studies have shown various MGO modifications on arginine residues of glycolytic enzymes; however, in our study, the majority of modified sites resided on lysine residues [12,24]. This may be due to treatment durations, as MG-H modifications occur more rapidly than carboxyethylations, but also have a shorter half-life. Interestingly, glycolysis is the major pathway responsible for the formation of MGO by spontaneous degradation of intermediate triosephosphates [4]. Therefore, this may suggest a potential regulatory feedback mechanism whereby elevated MGO leads to modification of glycolytic proteins, decreasing their activity and lowering MGO back to a physiological concentration. Modification of fructose bisphosphate aldolase A at K229 or 146 was identified in all cell types analysed in this study, including PBL. Remarkably, all four modification sites of ALDOA were involved in its aldolase activity [28]. K229 acts as a nucleophile in the formation of the Schiff-base with the C2-carbonyl group of the substrate fructose 1,6-bisphosphate, followed by cleavage of the C3-C4 bond to release the glycolytic intermediates G3P and DHAP [28]. The charged form of K146 non-covalently interacts with the substrate and intermediates to stabilise negative charges [28]. K41 and R42 have also been shown to interact with the 6-phosphate of the substrate [29]. All these interactions rely on the positively charged nucleophilic groups of the lysine or arginine residues which are lost following MGO modifications [30,31]. Therefore, it is highly likely that MGO modification of ALDOA at these residues, in particular K229, will affect substrate binding and its catalytic activity to interfere the progression of glycolysis. Moreover, modification of ENO1 was also shown to be a target of MGO, where seven modifications sites were detected following MGO treatment. Regulation of ENO1 activity has recently been shown to occur, at least in part by p300 mediated 2-hydroxyisobutyrylation of K228 [32], which was a site modified by MGO in our study. Knockdown of p300, which reduced the level of K228 2-hydroxyisobutyrylation, was shown to lower enzyme activity and the concentration of glycolytic intermediates [32]. Therefore, MGO modification of K228 could prevent p300 mediated 2-hydroxyisobutyrylation and cause a reduction in ENO1 glycolytic activity. MGO has been shown to decrease the activity of glycolytic enzymes in vitro and alter the abundance of glycolytic intermediates, specifically an increase in early stage (fructose 1,6 bisphosphate, dihydroxyacetone phosphate, 3-phosphoglycerate) and a decrease in later stage intermediates (phosphoenolpyruvate and pyruvate) [33,34,35,36]. Unfortunately, most studies that have investigated the activity of specific glycolytic enzymes in response to MGO have used high concentrations (2.5–5 mM) that increase the likelihood of non-specific MGO modifications [33,34,35,36]. Therefore, further studies investigating the effects of site-specific MGO modifications on the activity of glycolytic enzymes (under normal or pathophysiological conditions) would provide valuable information and reveal clues to the mechanistic action displayed by MGO. A recent study showed that treatment of GLO2 knockout cells with 50 µM MGO was able to decrease metabolic output through modification of the glycolytic enzymes by the glyoxalase intermediate S-D-lactoylglutathione, which produces lact(o)yllysine [37]. Interestingly, lact(o)yllysine and CEL are isomeric and may have similar signalling consequences. Both modifications produce identical mass shifts (+72) and can be difficult to differentiate by standard proteomics techniques. Nevertheless, both can be produced either directly (CEL) or indirectly (lact(o)yllysine) by MGO and therefore are associated with dicarbonyl stress.

Other cellular processes over-represented with MGO modifications included the spliceosome and ribosome. The spliceosome was previously shown to be enriched with MGO modifications and associated with a decrease in the abundance of proteins in this pathway [14]. The authors showed, using the Cancer Cell Line Encyclopedia (CCLE) that GLO1 mRNA expression positively correlated with the expression of spliceosome proteins (PPIL1 and CDC5L), suggesting increased GLO1 expression protects spliceosome proteins in tumour cells [14]. The spliceosome protein family serine and arginine-rich splicing factors (SRs), including SRSF2, which was modified with MGO in our study, are master regulators of pre-mRNA splicing. Dysregulation of this process can cause genomic instability and impede the normal expression pattern of proteins, resulting in aberrant biological function [38]. Loss of SRSF2 in mouse embryo fibroblasts cause cell cycle arrest in G2/M and double-strand break formation due in at least part to hyperphosphorylation and hyperacetylation of p53 [39]. Furthermore, mutations in SRSF2 and SF3B1, which also contained MGO modifications in this study, are observed in 5–75% of patients with various myeloid neoplasms, particularly chronic myelomonocytic leukemia and refractory anaemia with ring sideroblasts [40]. Therefore, protection of spliceosome pathway proteins may confer another role of GLO1 in tumour cells. 

Our study reveals glycolysis/gluconeogenesis, spliceosome and ribosome as major STRING pathways enriched with MGO modifications. Although we were able to identify many MGO modified proteins, it is unlikely to be a comprehensive list. Further efforts to enrich those proteins which are modified by MGO are required, particularly in the identification of low abundant proteins that may harbour these modifications. Nevertheless, our study provided valuable insights into the potential impact MGO may have on cellular function. Furthermore, site-specific identification of MGO modification sites suggests they may have a role in regulating glycolytic output by altering the activity of glycolytic enzymes. 

## 4. Materials and Methods

### 4.1. Materials

All reagents, chemicals and enzymes were purchased from Sigma unless indicated otherwise. Isotopically labelled and unlabelled MG-H1, CEL and Lysine were purchased from Iris Biotech (Marktredwitz, Germany). Trypsin Gold (Promega, V5280) and ProAlanase (Promega, VA2161) were purchased from Promega (Madison, WI, USA). Methylglyoxal solution was purchased from Sigma (St. Louis, MO, USA; M0252; Batch number BCBL7249V).

### 4.2. WIL2-NS Cell Culture

WIL2-NS cells (ATCC CRL-8155) was kindly gifted by Commonwealth Scientific Research Organisation (Adelaide, Australia). WIL2-NS cells were cultured in complete RPMI-1640 medium supplemented with 5% (*v*/*v*) fetal calf serum (FCS), l-glutamine (1% *v*/*v*) and penicillin/streptomycin (1% *v*/*v*) at 37 °C in a humidified atmosphere with 5% CO_2_. Cells were seeded at 5 × 10^5^ cells/mL and incubated for 24 h before being treated with 500 µmol/L MGO for a further 24 h.

### 4.3. Whole-Cell Quantification of MG-H1 and CEL

WIL2-NS cells were lysed in ice-cold RIPA buffer with sonication over ice for two 10 s bursts (Misonix, NY, USA). Cellular debris was removed by centrifugation (17,000× *g* for 10 min at RT). Protein was precipitated by the addition of ice-cold acetone (4:1 ratio of acetone: sample). Sample was left overnight at −20 °C before being centrifuged at 17,000× *g* for 10 min at RT. Precipitated protein was washed twice with ice-cold acetone before being resuspended in 50 mM ammonium bicarbonate (pH 8). Samples were incubated at 37 °C for 4 h to aid protein solubilization. Undissolved protein was removed by centrifugation at 17,000× *g* for 10 min (RT) and the protein concentration quantified using the Bicinchoninic acid (BCA; Sigma) assay following the manufacturer’s instructions. Trypsin (TPCK treated, ≥10,000 BAEE units/mg protein) was added at a enzyme to protein ratio (1:50) and the sample incubated at 37 °C for 16 h. Following incubation, sample was heated to 95 °C for 10 min to denature the trypsin. Following this, sample was cooled to room temperature and Pronase E (≥3.5 units/mg protein) and Aminopeptidase (≥12 units/mg protein) were added at protein-to-enzyme ratio (1:50) for a further 24 h. Enzymes were removed by the addition of ice-cold acetone (4:1 ratio of acetone: sample) and the sample centrifuged at 17,000× *g* for 10 min at RT. Supernatant was collected, dried under vacuum centrifugation and resuspended in 0.1% formic acid (*v*/*v*) containing the internal standard (100 nmol/L). Sample (2 μL) was injected and analytes were separated using a 150 × 4.6 mm, 4 µm Phenomenex C18 column (Phenomenex, Torrance, CA, USA) with a linear gradient of 0.1% formic acid in water (Buffer A) and 0.1% formic acid in acetonitrile (Buffer B) over 5 min at a flow rate of 0.6 mL/min. Multiple reaction monitoring (MRM) was conducted in positive mode using an AB Sciex 6500+ QTRAP mass spectrometer (Framingham, MA, USA) with the following transitions: *m*/*z* 147.4 > 83.9 (lysine), 151.2 > 87.9 (d_4_ lysine), 219.2 > 130.2 (CEL), 222.2 > 134.2 (d_4_CEL), 229.2 > 116.1 (MG-H1) 232.2 > 116.1 (d_3_ MG-H1). The ion source parameters were as follows: source temperature (450 °C), curtain gas (20 psi), collision gas (medium), ion spray voltage (5500 V) and ion source gas 1 and 2 (40 psi). The concentration of MG-H1 and CEL was normalized to lysine content and expressed as mmol analyte/mol lysine. Concentration of MG-H1 and CEL are expressed as mean ± SD (*n* = 3). Student *t*-test was conducted to determine any significant difference (*p* < 0.05) using GraphPad Prism (San Diego, CA, USA, Version 8.3.0).

### 4.4. Isolation and Purification of Peripheral Blood Lymphocytes

Venous blood was collected and lymphocytes isolated as previously described from three healthy male volunteers, aged 25–45 years by density gradient centrifugation [41]. Following isolation, PBL were stored in liquid nitrogen. The study was approved by the Human Research Ethics Committee of University of South Australia (Application ID: 203348) and written informed consent was obtained from all participants. For PBL, only cytoplasmic proteins were isolated by separation of cytoplasmic fraction from nuclear fraction by cell lysis with buffer containing 320 mM sucrose, 10 mM Tris-HCl, 5 mM MgCl_2_ and 1% Triton-X100. Supernatant was collected and protein precipitated with three volumes of ice-cold acetone and washed three times. Protein was dissolved in 8 M urea/50 mM ammonium bicarbonate and the protein concentration determined by BCA assay. 

### 4.5. Proteomic Analysis of WIL2-NS and PBL

Cell lysis for and protein isolation for WIL2-NS was identical to whole-cell quantification of MG-H1 and CEL; however, following this, protein was resuspended in 50 mM ammonium bicarbonate/8 M urea. Protein concentration was measured by BCA assay. Preparation of WIL2-NS and PBL for proteomic analysis was identical, except that PBL was only digested with trypsin. Briefly, 100 µg of protein was reduced and alkylated with 10 mM dithiothreitol and 15 mM chloroacetamide, respectively. Sample was diluted with 50 mM ammonium bicarbonate to reduce urea concentration to 0.8 M. Trypsin (Trypsin Gold, Promega, V5280, 2 µg) was added and samples were incubated for 8 h at 37 °C at 500 rpm. Following incubation, 10.2 µL of formic acid was added to terminate the reaction and acidify the sample. For ProAla digestion, 20 µg of protein was reduced and alkylated with 10 mM dithiothreitol and 15 mM chloroacetamide, respectively. Sample was diluted with 33 mM HCl to reduce the urea concentration to 0.3 M and pH to ~1.5. ProAlanase enzyme (Promega, VA2161, 0.2 µg) was added, and samples were incubated for 4 h at 37 °C. All subsequent steps were the same for trypsin and ProAla digests. Peptides were purified using in-house packed C18 stage tips [42]. Stage tips were conditioned with 40 µL of acetonitrile and equilibrated with 80 µL of 0.1% (*v*/*v*) formic acid. Samples were loaded onto the stage tip and washed with 40 µL of 0.1% (*v*/*v*) formic acid. Samples were then eluted with 50% (*v*/*v*) acetonitrile in 0.1% (*v*/*v*) formic acid. Eluates were dried using a Speed-Vac and reconstituted in 20 µL of 0.1% (*v*/*v*) formic acid. Peptide concentration was determined by measuring tryptophan concentration of the peptides using an Agilent Cary Eclipse Fluorescence Spectrophotometer. Briefly, samples diluted 1:10 in 8 M urea/50 mM ammonium bicarbonate were loaded into a 50 µL Quartz cuvette and fluorescence was measured at excitation and emission of 295 nm and 350 nm, respectively. Concentration was determined by comparing against a standard curve generated using tryptophan. Protein concentration was deduced from tryptophan concentration assuming a tryptophan content of 1.17% in mammalian proteins [43]. 

OV90 and Caov3 cell lines were prepared as described above except that a mixture of Trypsin/Lys-C was used for the digestion and samples incubated overnight. These samples were prepared for another study, but raw data files were processed as described below for analysis of MGO modifications.

### 4.6. High-Resolution Orbitrap Mass Spectrometry

LC-MS analysis was conducted on an EASY-nLC 1200 system coupled to an Orbitrap Exploris 480 mass spectrometer (Thermo Scientific, Bremen, Germany). Peptides from trypsin (1 µg) or ProAla (0.5 µg) digests were reconstituted in 0.1% formic acid and loaded onto a 25 cm fused silica column (75 µm inner diameter, 360 µm outer diameter) heated to 50 °C. The column was packed in-house with 1.9 µm ReproSil-Pur 120 C18-AQ particles (Dr. Maisch, Ammerbuch, Germany). Peptides were separated over a 70-min linear gradient (3 to 20% acetonitrile in 0.1% formic acid) at a flow rate of 300 nL/min. A FAIMS Pro interface (Thermo Scientific) generated compensation voltages at −50 and −70 V to regulate the entry of ionized peptides into the mass spectrometer. MS scans (*m*/*z* 300 to 1500) were acquired at resolution 60,000 (*m*/*z* 200) in positive ion mode. Peptide fragmentation (minimum threshold of 1 × 10^4^ precursor ions) was performed with 27.5% HCD collision energy, with the resulting MS/MS scans (starting *m*/*z* 85) measured at resolution 15,000. Cycle time was limited to 1.5 s, while the dynamic exclusion period was specified at 40 s.

### 4.7. Data Analysis

Raw MS files were analysed using Proteome Discoverer 2.4 (Thermo Scientific, Bremen, Germany). Data were processed with the Sequest HT search engine against a concatenated database containing the 74,811 forward entries from the UniProt human database (1 December 2019) and their respective decoy counterparts. Trypsin or ProAlanase (cleaves C-terminal to proline and alanine residues) was specified as the enzyme and a maximum of five miscleavages was allowed. Precursor and fragment mass tolerances were 10 ppm and 0.02 Da, respectively. Arginine and lysine carboxyethylation (+72.021129), arginine methylglyoxal-hydroimidazolone (+54.010565), methionine oxidation (+15.994915), N-terminal acetylation (+42.010565), N-terminal methionine loss (−131.040485), N-terminal methionine loss and acetylation (−89.02992) were set as dynamic modifications, with cysteine carbamidomethylation (+57.021464) designated a static modification. Protein and peptide identification false discovery rates were both set at 1%.

Enrichment analysis was performed using the Database for Annotation, Visualization and Integrated Discovery (DAVID) v6.8 (https://david.ncifcrf.gov/) (accessed 9 December 2021) [19]. The STRING database v11.5 (Search Tool for the Retrieval of Interacting Genes, available at: http://string-db.org/, accessed 5 January 2022 [19]) was used for construction of the protein–protein interaction network. Only interactions with an interaction score greater than 0.95 were included and any disconnected nodes were removed. Remaining STRING network was exported to Cytoscape [20]. Cluster labels were assigned based on the KEGG analyses of pathways that were associated with the greatest number of proteins in that cluster. Venn diagrams were generated using https://bioinformatics.psb.ugent.be/webtools/Venn/, accessed 8 November 2021.

## Figures and Tables

**Figure 1 ijms-23-03689-f001:**
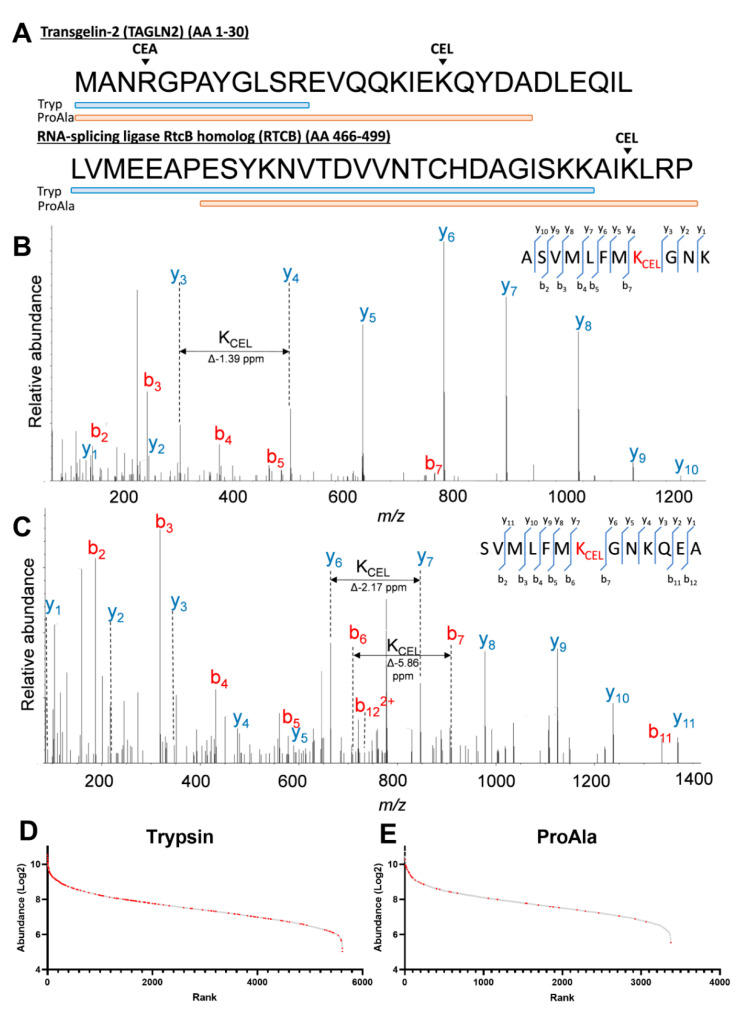
(**A**) Sequence coverage plot of Transgelin-2 (top) and RNA-splicing ligase RtcB homolog (bottom) protein sequences obtained from trypsin (blue) and ProAla (orange). (**B**,**C**) MS/MS spectra of a CEL modification on Glutaredoxin at K253 generated by trypsin digestion (**B**) and ProAla digestion (**C**) (see also Appendix A). (**D**,**E**) Abundance plot of proteins in trypsin digestion (**D**) and ProAla digestion (**E**) by rank. Grey spots represent unmodified proteins and red are those with MGO modifications.

**Figure 2 ijms-23-03689-f002:**
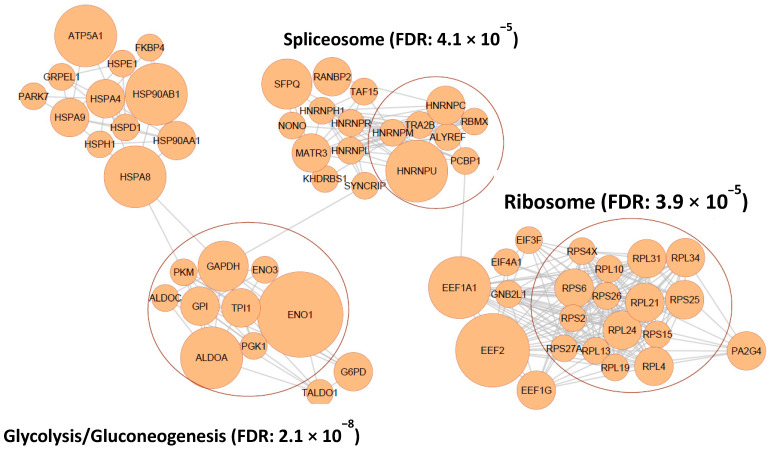
Protein–protein interaction network of MGO-modified proteins generated using STRING database v11.5 and Cytoscape v3.9.0 [19,20]. Size of the node (protein) reflects the number of MGO modifications found on that protein (ranging from 1–7). FDR; false discovery rate.

**Figure 3 ijms-23-03689-f003:**
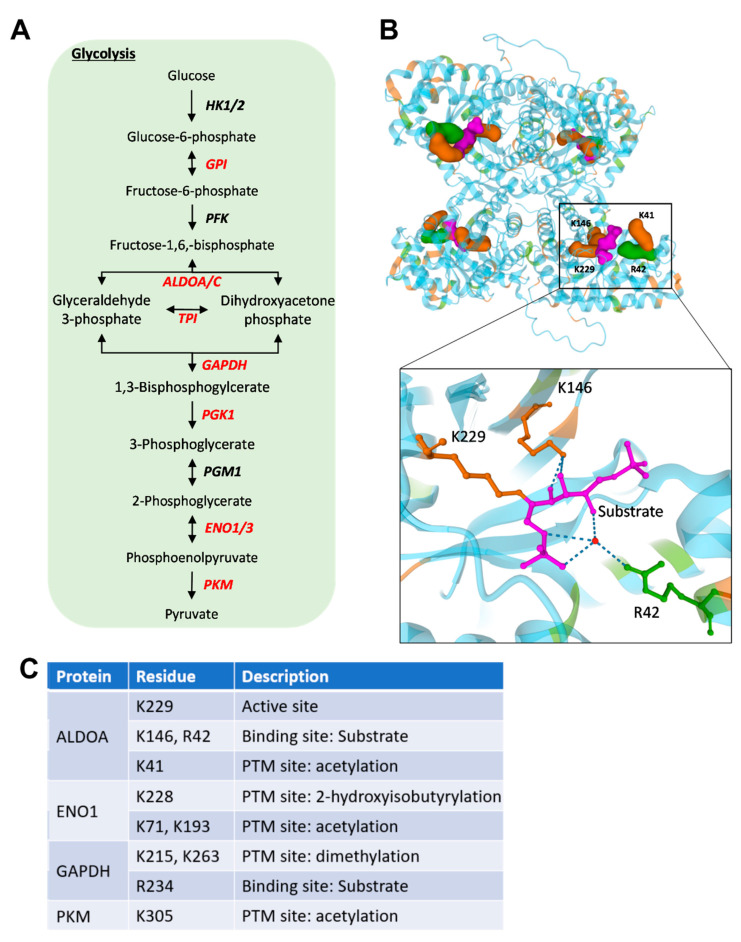
(**A**) Flow diagram showing glycolysis. Enzymes highlighted in red contain MGO modifications detected by LC-MS analysis. (**B**) Three-dimensional structure of fructose bisphosphate aldolase A (ALDOA, PDB entry 1ZAI). MGO-modified residues found in this study are labelled. and relevant arginine (green) and lysine (orange) residues are shown. Active site bound to intermediate Schiff-base substrate (purple) is enlarged. (**C**) Table of MGO modification sites that occur on residues involved in protein function. Features of each residue was obtained from UniProt. HK; hexokinase, GPI glucose-6-phosphate, PFK; phosphofructokinase, ALDO; fructose bisphosphate aldolase, TPI; triosephosphate isomerase, GAPDH; glyceraldehyde 3-phosphate dehydrogenase, PGK; phosphoglycerate kinase, PGM; phosphoglucomutase, ENO; enolase, PKM; pyruvate kinase M.

**Figure 4 ijms-23-03689-f004:**
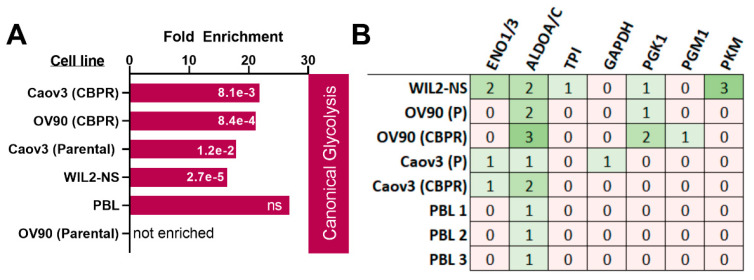
(**A**) Enrichment of MGO-modified glycolytic proteins in different cell types. For this analysis, all cells were untreated. For PBL, modified proteins from all three donors were collated into one list and enrichment analysis subsequently performed. FDR values are in white. (**B**) Occurrence of MGO-modified glycolytic enzymes in different cell types. CBPR; chemoresistance to the chemotherapeutic drug carboplatin, PBL; peripheral blood lymphocytes, ENO; enolase, ALDO; fructose bisphosphate aldolase, TPI; triosephosphate isomerase, GAPDH; glyceraldehyde 3-phosphate dehydrogenase, PGK; phosphoglycerate kinase, PGM phosphoglucomutase, PKM; pyruvate kinase M, ns; not significant.

**Table 1 ijms-23-03689-t001:** Enrichment analysis for biological processes enriched with MGO modifications.

Biological Process	Count	Fold Enrichment	FDR	Genes
**Canonical Glycolysis**	9	13.4	3.6 × 10^−5^	**ALDOA**, **ALDOC**, **ENO1**, **ENO3**, GPI, GAPDH, **PGK1**, **PKM**, **TPI**
Gluconeogenesis	9	7.9	1.6 × 10^−3^	**ALDOA**, **ALDOC**, **ENO1**, **ENO3**, GPI, GOT2, GAPDH, **PGK1**, **TPI**
**Translation initiation**	22	6.2	2.3 × 10^−8^	**DDX3Y**, DHX29, EIF3F, EIF4A1, **PABPC1**, PAIP1, RPL10, **RPL13**, **RPL19**, **RPL21**, RPL24, **RPL31**, **RPL34**, **RPL4**, RPS15, RPS2, RPS25, RPS26, RPS27A, **RPS4X**, **RPS4Y1**, **RPS6**
**Regulation of cellular response to heat**	12	6.2	4.7 × 10^−4^	FKBP4, HSP90AA1, **HSP90AB1**, **HSPA8**, HSPH1, **NUP153**, NUPL2, **POM121C**, **POM121**, **RANBP2**, **RPA3**, **YWHAE**
Nucleosome assembly	17	5.5	1.4 × 10^−5^	ATRX, H2AFX, ANP32A, **ANP32B**, DAXX, **HMGB2**, **HIST1H1A**, HIST1H2BB, HIST1H2BD, **HIST1H2BK**, HIST1H2BM, HIST1H3A, **HIST1H4I**, HIST2H2BF, HIST2H3PS2, **H3F3A**, NPM1
**Cell-cell adhesion**	38	5.4	1.3 × 10^−13^	**ALDOA**, **AHNAK**, CCT8, **CKAP5**, DDX6, DHX29, EEF1G, EEF2, **EHD4**, **ENO1**, FASN, FLNB, LASP1, **HSP90AB1**, **HSPA8**, HDLBP, HIST1H3A, HCFC1, LDHA, **PARK7**, **PRDX1**, **PCBP1**, PSMB6, **PPME1**, **PKM**, **RAB1A**, RANGAP1, RACK1, RPL24, **RPL34**, RPS2, RPS26, **SPTAN1**, SPTBN1, TAGLN2, **YWHAB**, **YWHAE**, **YWHAZ**
**mRNA splicing, via spliceosome**	26	4.5	2.4 × 10^−7^	ALYREF, **CSTF2**, **DNAJC8**, **FUS**, **HSPA8**, **HNRNPC**, **HNRNPH1**, **HNRNPH2**, **HNRNPL**, HNRNPM, HNRNPR, **HNRNPU**, NONO, **PPIE**, **PABPC1**, **PCBP1**, **POLR2B**, **RBMX2**, **RBMX**, **SRSF2**, **SNRPB**, **SPEN**, **SF3B1**, **SF3B2**, SYNCRIP, TRA2B
Protein folding	21	4.5	9.9 × 10^−6^	**CANX**, **CCT2**, CCT5, CCT8, FKBP4, FKBP5, GRPEL1, HSPE1-MOB4, HSP90AA1, **HSP90AA2P**, **HSP90AB1**, HSP90AB2P, **HSP90B1**, **HSPA8**, HSPA9, HSPE1, PPIA, **PPIE**, **RANBP2**, ST13, TXN
**Protein sumoylation**	13	4.3	5.2 × 10^−3^	**BIRC5**, **HNRNPC**, IFIH1, **NUP153**, NUPL2, **POM121C**, **POM121**, **RAD21**, **RANBP2**, **RING1**, **SMC3**, **TRIM28**
G2/M transition of mitotic cell cycle	15	4.2	1.5 × 10^−3^	ALMS1, **BIRC5**, CNTRL, **CEP250**, **CKAP5**, **HAUS5**, HSP90AA1, **KHDRBS1**, ODF2, PPP1CB, RPS27A, **TUBA4A**, TUBB4B, **YWHAE**

Pathway enrichment analysis was performed using Database for Annotation, Visualization and Integrated Discovery (DAVID) v6.8 (https://david.ncifcrf.gov/, accessed on 9 December 2021) [21]. Biological Processes or proteins in **bold** were also observed in control (untreated) cells.

## Data Availability

Raw MS files will be uploaded to a publicly available repository upon acceptance of the manuscript.

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
