# Peer review of "Proteomic Analysis of Methylglyoxal Modifications Reveals Susceptibility of Glycolytic Enzymes to Dicarbonyl Stress"

_ijms, 2022, doi:10.3390/ijms23073689_

Round 1

Reviewer 1 Report

21 February 2022

Journal: International journal of molecular sciences, MPDI

Title:  Proteomic analysis of methylglyoxal modifications reveals susceptibility of glycolytic enzymes to dicarbonyl stress

Authors:  Leigh Donnellan, et al.

Summary:

The authors report methylglyoxal (MGO)-protein modifications in WIL2-NS, OV90, CaoV3 cell lines, and peripheral blood lymphocytes (PBL) using exogenous MGO. A total of 520 MGO-modified proteins were found.   Glycolytic proteins were enriched. The data are reported in 4 figures and 1 table.  Supplementary information contains Appendix A (two figures and one table), Appendix B (An excel sheet containing 6 tabs, that catalog the MGO-modified proteins in different cell lines), Appendix C reports the MS2 spectra of the MGO modified glycolytic enzymes. Identification of the MGO modified proteins during the dicarbonyl stress is important and helps in understanding the complications associated with it. The following critiques are provided to help improve the clarity and scientific content of the manuscript.

Critique:

  1. The figure legends need major revisions
    1. The figure titles should be a short description, not a paragraph-long sentence that describes the methods used.
    2. “Figure 1” on page 7 is mislabeled. This should be “Figure 3”.
    3. “Figure 2” on page 8 is mislabeled. This should be “Figure 4”.
    4. Abbreviations should be included in the figure legend. For example, the abbreviation CBPR is used in Figure 4 on page 8, but the reader is left guessing what this means.  After some effort, it was apparent this was an abbreviation for carboplatin-resistant.  There is no definition and no mention of the level of carboplatin resistance, or if the cells labeled CBPR are routinely grown in a specific concentration of carboplatin to limit loss of resistance.
  2. 520 MGO-modified proteins were identified in the WIL2-NS cell lines, but the Appendix B (WIL2-NS) showed only 519 MGO-modified proteins. On the one hand, this is a minor issue. On the other hand, it raises the possibility of a cut-and-paste error, or other systematic error.
  3. The authors identified 7 of 10 glycolysis enzymes with MGO modifications. Is there any functional data that the activity of these enzymes was changed?
    1. Methylglyoxal is chemically derived by the reduction of pyruvate from a carboxylic acid to an aldehyde. Were there metabolic changes in the concentration of glucose, pyruvate, or any intermediates of glycolysis?
  4. Appendix B, tab caov2 (parental), the Score Sequest HT values, MG-H1 and carboxyethyl positions of the modified proteins were missing.
  5. Peripheral blood lymphocytes (PBL) collected from 3 male volunteers were presented in the study. But the demographics of the patients are not shown (age, glucose levels, etc).
  6. It is not clear if the PBLs were untreated or treated with exogenous MGO.
  7. Mass spectrometry parameters important for replication were not reported. These parameters include polarity, spray voltage, source temperature, curtain gas psi, etc.
  8. Line 147. It is customary to write out the full name of the enzymes when it is used for the first time in the manuscript along with the abbreviations, in the subsequent discussion the abbreviations are fine. None of the full names of the enzymes were given.
  9. Line 171. What is “Figure 2. NS”?

Typos

  1. Line 117- Useful fo  useful of
  2. Line 137, 139, 144. Splicesome – spliceosome. Spliceosome spelling should be corrected at several places in the manuscript.
  3. Line 159. Despite the presence of of – Despite the presence of
  4. Line 205. Seevral – Several
  5. Line 213. Permorformed – performed.
  6. Line 237. Digestiong – Digestion
  7. Appendix C: “(uniport R43)” should be “(uniprot…)”

Author Response

Reviewer comment 1: The figure legends need major revisions

    1. The figure titles should be a short description, not a paragraph-long sentence that describes the methods used.
    2. “Figure 1” on page 7 is mislabeled. This should be “Figure 3”.
    3. “Figure 2” on page 8 is mislabeled. This should be “Figure 4”.
    4. Abbreviations should be included in the figure legend. For example, the abbreviation CBPR is used in Figure 4 on page 8, but the reader is left guessing what this means.  After some effort, it was apparent this was an abbreviation for carboplatin-resistant.  There is no definition and no mention of the level of carboplatin resistance, or if the cells labeled CBPR are routinely grown in a specific concentration of carboplatin to limit loss of resistance.

Response: We thank the reviewer for these constructive comments relating to the figure legends. We have revised each figure legend by removing any unnecessary information. Furthermore, as suggested by the reviewer, to improve clarity of each figure we have included in the figure caption all abbreviations used in the corresponding figure. We believe issues relating to the mislabelling (comment 1(2 and 3)) were the result of incorrect formatting prior to submission. Correct formatting has now been applied and the issue should be rectified.   

Reviewer comment 2: 520 MGO-modified proteins were identified in the WIL2-NS cell lines, but the Appendix B (WIL2-NS) showed only 519 MGO-modified proteins. On the one hand, this is a minor issue. On the other hand, it raises the possibility of a cut-and-paste error, or other systematic error.

Response: We thank the reviewer for picking up on this oversight. The correct number of MGO-modified proteins is 519, rather than 520. We believe this occurred by not excluding the column titles when counting number of rows (proteins) in the list while combining all MGO modified proteins. To further confirm this number, we cross-checked against Proteome Discoverer files and the number of modified proteins is 519.    

Reviewer comment 3: The authors identified 7 of 10 glycolysis enzymes with MGO modifications. Is there any functional data that the activity of these enzymes was changed?

    1. Methylglyoxal is chemically derived by the reduction of pyruvate from a carboxylic acid to an aldehyde. Were there metabolic changes in the concentration of glucose, pyruvate, or any intermediates of glycolysis?

Response: Unfortunately, as part of this study there is no functional data relating to changes in glycolytic output due to MGO. The purpose of this manuscript was to identify the pathways that may be most susceptible to elevated MGO. Having now identified glycolysis as one such pathway, the functional impact of MGO on this pathway will be under further investigation.

Reviewer comment 4: Appendix B, tab caov2 (parental), the Score Sequest HT values, MG-H1 and carboxyethyl positions of the modified proteins were missing.

Response: We apologise for this missing information as a result of formatting error. The Sequest HT values, MG-H1 and CEL positions were present in the file, however, the column for Sequest HT was extended out of view, such that all values were hidden. The re-formatted file is now uploaded to be consistent with other tabs.

Reviewer comment 5: Peripheral blood lymphocytes (PBL) collected from 3 male volunteers were presented in the study. But the demographics of the patients are not shown (age, glucose levels, etc).

Response: Peripheral blood lymphocytes (PBL) used in this study were initially harvested for a separate manuscript in which they were treated with methylglyoxal (MGO) to measure its genotoxic effect (1). Due to the design of the original study, parameters such as glucose levels were not determined. However, age range of the participants has now been included in the revised manuscript. Nevertheless, we were primarily interested in MGO modifications sites present in all samples, rather than those only appearing in individuals with high glucose levels or other metabolic abnormalities.  

Reviewer comment 6: It is not clear if the PBLs were untreated or treated with exogenous MGO.

Response: To provide clarity over treatment conditions of epithelial cell lines and PBL the following sentences have been included in the revised manuscript. “Therefore, using previous data obtained from epithelial cell lines; OV90 and Caov3 without (parental) and with chemo-resistance to the chemotherapeutic drug carboplatin (CBPR) from our laboratory, raw data files were reprocessed for MGO modifications. As these cell lines were initially used for a different study, they were not treated with MGO and therefore modification sites represent those present under basal conditions. All cell lines, except for OV90 (parental) showed canonical glycolysis to be over-represented with MGO modifications (Figure 4A). Next, using peripheral blood lymphocytes isolated from three separate healthy male donors, we investigated whether the same was true in vivo. As with epithelial cell lines, analysis of MGO modification sites was done without MGO treatment to determine modifications sites present under basal conditions only.”

Reviewer comment 7: Mass spectrometry parameters important for replication were not reported. These parameters include polarity, spray voltage, source temperature, curtain gas psi, etc.

Response: Ion source parameters are now included in section 4.3 Whole cell quantification of MG-H1 and CEL...” The ion source parameters were as follows; source temperature (450°C), curtain gas (20 psi), collision gas (medium), ion spray voltage (5500 V) and ion source gas 1 and 2 (40 psi).”

Reviewer comment 8: Line 147. It is customary to write out the full name of the enzymes when it is used for the first time in the manuscript along with the abbreviations, in the subsequent discussion the abbreviations are fine. None of the full names of the enzymes were given.

Response: We thank the reviewer for pointing this out and have updated the text to provide all full names… ‘Remarkably, 7 of the 10 steps contained MGO-modified proteins including glucose-6-phosphate isomerase (GPI), enolase 1/3 (ENO1/3), fructose bisphosphate aldolase A/C (ALDOA/C), triosephosphate isomerase (TPI), glyceraldehyde 3-phosphate dehydrogenase (GAPDH), phosphoglycerate kinase 1 (PGK1) and pyruvate kinase M (PKM) consisting of 20 modification sites in total (Figure 3A and Appendix C).’

Reviewer comment 9: Line 171. What is “Figure 2. NS”?

Response: It appears to be a formatting error in the submitted files as one line was erased from the text. This has been fixed and the sentence now reads as “Following the identification of glycolytic enzymes as primary targets for MGO-modification in WIL2-NS we sought…”

  1. Donnellan, L., Simpson, B., Dhillon, V.S., Costabile, M., Fenech, M., and Deo, P. (2021) Methylglyoxal induces chromosomal instability and mitotic dysfunction in lymphocytes. Mutagenesis.

Reviewer 2 Report

Dear authors,

you analysed several cell culture models via proteomics for methylglyoxal modifications after treatment with extracellular given MGO. Focus is on WIL2-NS B cells, and other cells have been analysed in parallel.

As a reviewer I am pretty fine with the manuscript, detailed information is given on procedure, detailled results are presented in the main text and in the supplements. Hopefully, the MGO is freshly distilled and not the purchased one from Sigma, that contains formalin and other by-products (this should be clarified).

Only criticism are slips of the pen, doubling in words which need to be eliminated (e.g. on page 8 Seevral, permorformed, on page 11 breifly etc.)

Author Response

Response: We thank the reviewer for their comments. Methylglyoxal (MGO) used in this study was purchased from Sigma (M0252: Batch number: BCBL7249V). This information has been added to the methods section of the revised manuscript. While commercial MGO has previously been shown to contain formalin, we believe that should not impact the results of this particular study as it was purely an analytical study, with no functional experiments involved. Therefore, we feel presence of formalin, or other contaminants should not affect the measurement of MGO modifications by high resolution orbitrap mass spectrometry. Nevertheless, the reviewer is correct in their concerns regarding commercial MGO, and subsequent studies aimed at investigating functional impact of MGO in glycolysis will require high purity MGO to prevent confounding effects of formalin or other contaminants.